applied mathematics/medical physics/ mathematical physics

integrals, series, neural field theory, Bessel functions, Green function, haemodynamic response function

**Author for correspondence:**
P. A. Robinson
e-mail: robinson@physics.usyd.edu.au

# Integrals and series related to propagators of neural and haemodynamic waves

## P. A. Robinson[1,2]

[1]School of Physics, and [2]Center for Integrative Brain Function, University of Sydney, Sydney, NSW 2006, Australia

PAR, 0000-0003-4100-6049

The propagator, or Green function, of a class of neural activity fields and of haemodynamic waves is evaluated exactly. The results enable a number of related integrals to be evaluated, along with series expansions of key results in terms of Bessel functions of the second kind. Connections to other related equations are also noted.

## 1. Introduction

Neural field theory (NFT) averages over the microscopic properties of neurons to obtain partial differential equations for mean levels of neural activity [1–6]. Propagation of activity across the cerebral cortex is governed by damped wave equations in NFT, whose Green functions, or propagators, embody the linear dynamics of such systems. In particular, the Green function represents the response evoked by a localized stimulus, which is a widely used method of probing brain function [7]. Such impulse responses have also been used to describe observed avalanche-like dynamics of neural activity [8,9].

Many useful cases of NFT propagators have a form that incorporates both damped wave propagation and regeneration of activity with a characteristic gain. Here, we analyse a particular form that applies to activity propagation across the two-dimensional (2D) surface of the cerebral cortex, approximating it as flat. The three-dimensional integral that must be carried out to obtain the propagator has not been evaluated in NFT, although we discuss below how it can be related to propagators found in relativistic quantum theory. Its direct evaluation, as done here, puts the results in a suitable form for use in NFT and enables a variety of other integrals to be evaluated that do not appear to be included in standard tables such as [10–14]. Notably, the propagator that governs slow haemodynamic disturbances in brain tissue is a limiting form of the one for neural activity, although the physical mechanisms described are quite different. Hence, the present work also allows the haemodynamic response function to be evaluated exactly [15].

**Figure 1.** Diagrammatic representation of the terms in equation (2.3), illustrating that $T$ is composed of local activation $I$ plus terms representing direct propagation and indirect propagation via multiple neural interactions. A factor of $g$ occurs at each vertex, representing regeneration of neural spikes, and the bare propagator $\Lambda$ carries activity between interactions.

In §2, we outline the necessary NFT background and the form of the propagator to be evaluated. Section 3 is devoted to evaluating the relevant integral exactly, while §4 explores a variety of other integrals and series that derive from it. Finally, §5 provides a summary.

# 2. Neural field theory propagators

This section briefly provides the background to the central integral on which the subsequent material is based. This integral has arisen in various contexts in models of neural activity and resulting haemodynamics, so the same notation is kept here. Moreover, the descriptions are kept as close as possible to those in the previous work cited here to avoid any ambiguity or confusion.

Neural activity propagates as voltage spikes in axons, and NFT deals with spiking activity averaged over many neurons to form a field at scales of millimetres and up [2]. In a broad class of NFTs, the propagation of moderate-amplitude local mean neural spiking-activity perturbations $\phi$ across the 2D cortical surface from a delta-function stimulus at $(\mathbf{r}, t) = (\mathbf{0}, 0)$ can be approximated as obeying the linear partial differential equation [1,2,6,7,16]

$$\left[\frac{1}{\gamma^2}\frac{\partial^2}{\partial t^2} + \frac{2}{\gamma}\frac{\partial}{\partial t} + 1 - \rho^2\nabla^2\right]\phi(\mathbf{r}, t) = \delta(\mathbf{r})\delta(t), \tag{2.1}$$

where $\mathbf{r}$ and $t$ denote position and time, $\rho$ is the characteristic range of the near-exponential distribution of axon lengths [1,4–6] and $\gamma$ is a characteristic damping rate. Equation (2.1) represents large-scale activity that propagates across the cortex as a damped wave at a speed $v = \gamma\rho$ [1,6], and can be expressed in integral form via its propagator [1,2,4,5].

Equation (2.1) describes direct propagation of activity without regeneration by neural interactions. If we approximate the cortical surface as being flat and neglect boundary conditions, a Fourier transform of equation (2.1) implies the direct propagator [1]

$$\Lambda(k, \omega) = \frac{1}{(1 - \mathrm{i}\omega/\gamma)^2 + k^2\rho^2}, \tag{2.2}$$

where $k$ is the wavenumber and $\omega$ the angular frequency. This propagator is represented by the arrow at upper right of figure 1. If there is a mean gain $g$ for regeneration of spikes at destination neurons, the outgoing activity at the next step is $g\Lambda$. Macroscopically, $g$ is the differential change in outgoing $\phi$ per unit change in incoming $\phi$ at each stage of post-stimulus neural interaction, with $g < 1$ for stability [1].

The total activity $T$ evoked by a delta input is the sum of local by the external stimulus, indicated by $I$ in figure 1, and the terms on the right of figure 1 weighted by corresponding powers of $g$. This gives [16,17]

$$T(k, \omega) = \sum_{n=0}^{\infty} [g\Lambda(k, \omega)]^n, \tag{2.3}$$

$$= [1 - g\Lambda(k, \omega)]^{-1}. \tag{2.4}$$

which converges for $g < 1$. Substitution of equation (2.2) into equation (2.4) gives

$$T'(k, \omega) = \frac{[T(k, \omega) - 1]}{g}, \tag{2.5}$$

$$= \frac{1}{(1 - i\omega/\gamma)^2 - g + k^2\rho^2}, \tag{2.6}$$

where $T'$ excludes local excitation (the $n = 0$ term in equation (2.3), which corresponds to the term $I$ in figure 1) and has been divided by $g$ for convenience; it represents the normalized propagating part of the response. From equation (2.6), it is evident that $T'$ satisfies equation (2.1), except for the appearance of $g$, which leads to the unit term inside the square brackets on the left being replaced by $1 - g$. Equations (2.3)–(2.5) exhibit the decomposition of neural propagation into the contributions of local activation ($n = 0$), direct propagation represented by $\Lambda$ ($n = 1$), and propagation that involves intermediate interactions ($n = 2, \ldots$). Therefore, $T(k, \omega)$ is the total propagator (or transfer function or Green function), and $\Lambda(k, \omega)$ is the direct propagator without regeneration [16], both of which depend on $\mathbf{k}$ only through its magnitude $k$.

If we take the limit $g \rightarrow 1$, we obtain a wave equation that has been used to describe the cortical haemodynamic responses to localized stimuli [15]. These responses are responsible for the blood oxygen level dependent (BOLD) signal that underlies functional magnetic resonance imaging (fMRI). In this context, the Green function obtained from equation (2.1) without the unit term within the large parentheses at left is termed the haemodynamic response function [15].

# 3. Evaluation of the propagator

Because the transfer function in equation (2.6) only depends on $k$, $T'$ is isotropic in 2D coordinate space. Fourier transformation of equation (2.6) then yields

$$T'(r, t) = \iint \frac{e^{i\mathbf{k}\cdot\mathbf{r} - i\omega t}}{(1 - i\omega/\gamma)^2 - g + k^2\rho^2} \frac{d^2\mathbf{k}}{(2\pi)^2} \frac{d\omega}{2\pi}, \tag{3.1}$$

$$= \int_0^\infty kJ_0(kr) \int \frac{e^{-i\omega t}}{(1 - i\omega/\gamma)^2 - g + k^2\rho^2} \frac{d\omega}{2\pi} \frac{dk}{2\pi}, \tag{3.2}$$

where $r = |\mathbf{r}|$. In all cases, causality requires that $T(r, t) = 0$ for $t < 0$, so we refer only to $t \geq 0$ in what follows. Likewise, we assume that integrals extend over their whole domain unless otherwise indicated.

If we perform the $k$ integral in equation (3.2), eqn 10.22.46 of [11] yields

$$T'(r, t) = \frac{1}{2\pi\rho^2} \int e^{-i\omega t} K_0\left[\frac{r}{\rho}\sqrt{\left(1 - \frac{i\omega}{\gamma}\right)^2 - g}\right] \frac{d\omega}{2\pi}, \tag{3.3}$$

where $K_0$ is a Macdonald function; i.e. a modified Bessel function of the second kind. We next write the argument of $K_0$ as $\lambda z$ with

$$z = \frac{r}{\rho}\left(1 - \frac{i\omega}{\gamma}\right) \tag{3.4}$$

and

$$\lambda = \left[1 - \frac{g}{(1 - i\omega/\gamma)^2}\right]^{1/2}, \tag{3.5}$$

so

$$\lambda^2 - 1 = \frac{-g}{(1 - i\omega/\gamma)^2}. \tag{3.6}$$

The multiplication theorem for Bessel functions of the second kind, or Macdonald functions, (eqn 10.44.1 of [11]) then implies

$$K_0(\lambda z) = \sum_{m=0}^\infty \frac{(-1)^m(\lambda^2 - 1)^m(z/2)^m}{m!} K_m(z), \tag{3.7}$$

which is valid for $|\lambda^2 - 1| < 1$, a condition that is satisfied for $g < 1$.

Hence,

$$T'(r, t) = \frac{1}{2\pi\rho^2} \int e^{-i\omega t} \sum_{m=0}^{\infty} \frac{1}{m!} \left[ \frac{gr}{2\rho(1 - i\omega/\gamma)} \right]^m K_m \left[ \frac{r}{\rho} \left( 1 - \frac{i\omega}{\gamma} \right) \right] \frac{d\omega}{2\pi}. \tag{3.8}$$

The identity

$$K_m(z) = \frac{\sqrt{\pi}}{\Gamma(m + (1/2))} \left( \frac{z}{2} \right)^m \int_1^{\infty} e^{-zu}(u^2 - 1)^{m-1/2} du, \tag{3.9}$$

(eqn 10.32.8 of [11]) can then be used to rewrite equation (3.8) as

$$T'(r, t) = \frac{1}{2\pi\rho^2} \sum_{m=0}^{\infty} \frac{\sqrt{\pi}}{m! \Gamma(m + (1/2))} \left( \frac{gr^2}{4\rho^2} \right)^m$$

$$\times \int_1^{\infty} e^{-ru/\rho}(u^2 - 1)^{m-1/2} \left[ \int \exp\left[ -i\omega\left( t - \frac{ru}{v} \right) \right] \frac{d\omega}{2\pi} \right] du, \tag{3.10}$$

with $v = \gamma\rho$. Hence,

$$T'(r, t) = \frac{1}{2\pi\rho^2} \sum_{m=0}^{\infty} \frac{\sqrt{\pi}}{m! \Gamma(m + (1/2))} \left( \frac{gr^2}{4\rho^2} \right)^m \int_1^{\infty} e^{-ru/\rho}(u^2 - 1)^{m-1/2} \delta\left( t - \frac{ru}{v} \right) du, \tag{3.11}$$

because

$$\int e^{-ibu} du = 2\pi\delta(b). \tag{3.12}$$

Using the property of the Dirac delta function that $\delta(ax) = \delta(x)/|a|$, and performing the $u$ integral, we then find

$$T'(r, t) = \frac{\gamma^2}{2\pi v} \frac{e^{-\gamma t}}{\sqrt{v^2 t^2 - r^2}} \sum_{m=0}^{\infty} \frac{\sqrt{\pi}}{m! \Gamma(m + (1/2))} \left( \frac{g}{4\rho^2} \right)^m (v^2 t^2 - r^2)^m, \tag{3.13}$$

for $r < vt$, with $T'(r, t) = 0$ otherwise because waves cannot reach $r > vt$ in time $t$.

We must now evaluate the series in equation (3.13), which has the form

$$S(y) = \sum_{m=0}^{\infty} \frac{y^m}{m! \Gamma(m + 1/2)} \tag{3.14}$$

and

$$y = \frac{g(v^2 t^2 - r^2)}{(4\rho^2)}. \tag{3.15}$$

Use of the identity

$$\Gamma\left( m + \frac{1}{2} \right) = \frac{(2m)!}{4^m m!} \pi^{1/2}, \tag{3.16}$$

then yields

$$S(y) = \pi^{-1/2} \cosh(2\sqrt{y}), \tag{3.17}$$

which has the same series expansion.

Hence, for $0 \le g \le 1$ and $r \le vt$, we obtain

$$T'(r, t) = \frac{\gamma^2}{2\pi v} \frac{e^{-\gamma t}}{\sqrt{v^2 t^2 - r^2}} \cosh\left[ \frac{\gamma\sqrt{g}}{v} \sqrt{v^2 t^2 - r^2} \right]. \tag{3.18}$$

At large $t$, the dominant term on the right side of equation (3.18) varies as $\exp[-\gamma(1 - \sqrt{g})t]$, which approaches zero as $t \to \infty$, given that we have assumed $g < 1$ for stability.

Figure 2 shows examples of the propagator in equation (3.18) at different times for $g = 0$, 0.5 and 0.9. For $g = 0$, we see a decaying activity front propagating outward at $v = \gamma\rho$. At higher $g$, the early evolution is very similar, but as waves are regenerated, the small-$r$ levels are higher and the overall decay is slower. These results improve on the approximate form recently obtained to illustrate $T'(r, t)$ in [8], although the scaling arguments in that paper are not affected by this refinement.

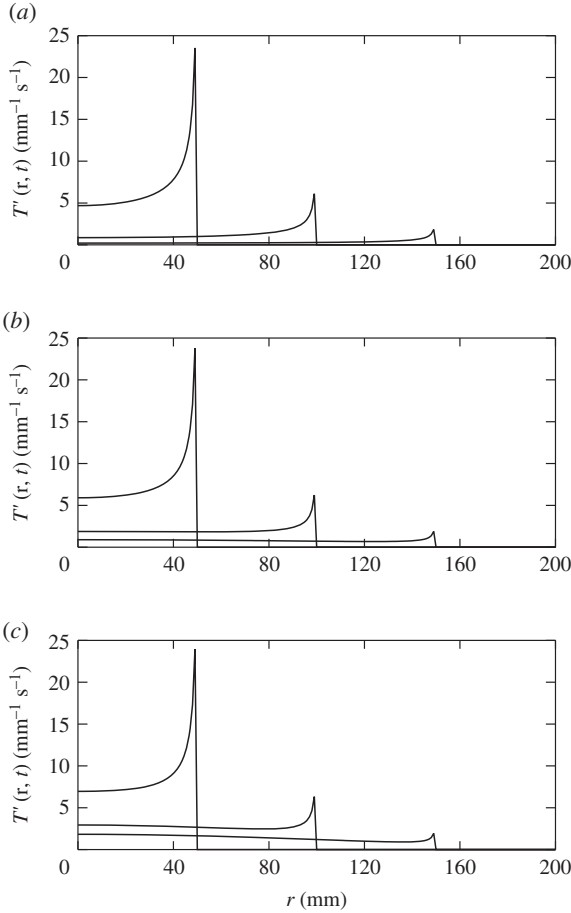

**Figure 2.** Propagator at various times for $\gamma = 200$ s$^{-1}$, $\rho = 0.05$ m and $v = \gamma\rho = 10$ m s$^{-1}$, showing $T'(r, t)$ versus $r$ from equation (3.18) for $t = 5, 10, 15$ ms, from left to right. (a) $g = 0$, (b) $g = 0.5$ and (c) $g = 0.9$.

It is worth noting that the modified version of equation (2.1) that includes $g$ is a special case of the 2D telegrapher's equation [18]. If one writes $\psi(\mathbf{r}, t) = e^{\gamma t}\phi(\mathbf{r}, t)$, $\psi$ obeys the Klein–Gordon equation for a scalar boson of mass $\sqrt{g}$, whose 2D solutions can be written in terms of Hankel functions [11,19–21]. If the advanced (past-propagating) part of the resulting propagator is discarded, the resulting retarded (future-propagating) propagator for $\psi$ can be shown to yield equation (3.18) for $\phi$. This approach, however, does not give the expressions for other integrals and series that we obtain via the present approach in this section and §4.

# 4. Related integrals and series

We can evaluate the $\omega$ integral in equation (3.2) by means of Cauchy's residue theorem, using a contour that follows the real axis and is closed clockwise by a semicircle in the lower half plane (the limit is taken in which the radius goes to infinity). This yields

$$T'(r, t) = \gamma^2 e^{-\gamma t} \int_0^\infty k J_0(kr) \frac{\sin\left[t\sqrt{k^2 v^2 - g\gamma^2}\right]}{\sqrt{k^2 v^2 - g\gamma^2}} \frac{dk}{2\pi}. \tag{4.1}$$

An extensive search of standard tables of integrals has not uncovered any expression for the integral in equation (4.1), but this result must be equal to the one in equation (3.18). Hence, we find

$$\int_0^\infty k J_0(kr) \frac{\sin\left[t\sqrt{k^2 v^2 - g\gamma^2}\right]}{\sqrt{k^2 v^2 - g\gamma^2}} dk = \frac{1}{v\sqrt{v^2 t^2 - r^2}} \cosh\left[\frac{\gamma\sqrt{g}}{v}\sqrt{v^2 t^2 - r^2}\right]. \tag{4.2}$$

Some special cases of the propagator $T'$ are as follows:

(i) To find the time-integrated activity that reaches a distance $r$, we integrate equation (3.2) over time, which is equivalent to extracting the $\omega = 0$ component. Thus, [1,10,11,16]

$$T'(r) = \int_0^\infty \frac{kJ_0(kr)}{k^2\rho^2 + 1 - g} \frac{dk}{2\pi}, \tag{4.3}$$

$$= \frac{1}{2\pi\rho^2} K_0\left[\frac{r}{\rho}\sqrt{1-g}\right]. \tag{4.4}$$

We can integrate equation (4.4) over $\mathbf{r}$ to obtain the total activity ultimately initiated by a unit impulse. This is equivalent to selecting the $(\mathbf{k}, \omega) = (\mathbf{0}, 0)$ component of $T(\mathbf{k}, \omega)$, which is $(1-g)^{-1}$.

The result in equation (4.4) must be equivalent to the integral of equation (3.18) over $t$ from $r/v$ to infinity. Hence,

$$\int_{r/v}^\infty \frac{e^{-\gamma t}}{\sqrt{v^2 t^2 - r^2}} \cosh\left[\frac{\sqrt{g}}{\rho}\sqrt{v^2 t^2 - r^2}\right] dt = \frac{1}{v} K_0\left[\frac{r}{\rho}\sqrt{1-g}\right], \tag{4.5}$$

or, equivalently,

$$\int_a^\infty \frac{e^{-\gamma t}}{\sqrt{t^2 - a^2}} \cosh\left[\gamma\sqrt{g}\sqrt{t^2 - a^2}\right] dt = K_0\left[\gamma a\sqrt{1-g}\right]. \tag{4.6}$$

(ii) If the total activity at a given time is sought, one must integrate equation (3.1) over position, which is equivalent to extracting the $\mathbf{k} = \mathbf{0}$ component. This gives

$$T'(t) = \int \frac{e^{-i\omega t}}{(1 - i\omega/\gamma)^2 - g} \frac{d\omega}{2\pi}, \tag{4.7}$$

$$= \frac{\gamma e^{-\gamma t}}{\sqrt{g}} \sinh(t\gamma\sqrt{g}). \tag{4.8}$$

This result can also be obtained by integrating equation (3.18) over $\mathbf{r}$ from the origin to $r = vt$. For $g < 1$, the right of equation (4.8) rises linearly before reaching a maximum and decaying as $\exp[-\gamma(1 - \sqrt{g})]$, whereas if $g = 1$ the initial linear rise is followed by an approach to the asymptotic constant level of $\gamma/(2\sqrt{g})$. For $g < 1$, the integral of equation (4.8) over time is $(1-g)^{-1}$ which is always greater than 1 because of regeneration of activity.

(iii) If $g = 0$, we find from equation (3.18)

$$T'(r, t) = \frac{\gamma^2 e^{-\gamma t}}{2\pi v\sqrt{v^2 t^2 - r^2}}, \tag{4.9}$$

which reproduces a result in [1]. This is also the expression for $\Lambda(r, t)$, as seen from equation (2.2).

A different family of integrals can be obtained by expanding the integrand in equation (3.2) in powers of $g$ for $g < 1$. This gives

$$T'(r, t) = \int\int \frac{e^{i\mathbf{k}\cdot\mathbf{r} - i\omega t}}{(1 - i\omega/\gamma)^2 - g + k^2 r^2} \frac{d^2\mathbf{k}}{(2\pi)^2} \frac{d\omega}{2\pi}, \tag{4.10}$$

$$= \sum_{m=0}^\infty g^m \int\int_0^\infty kJ_0(kr) \frac{e^{-i\omega t}}{[(1 - i\omega/\gamma)^2 + k^2\rho^2]^{m+1}} \frac{dk}{2\pi} \frac{d\omega}{2\pi}, \tag{4.11}$$

$$= \sum_{m=0}^\infty \frac{g^m}{\rho^2} \left(\frac{r}{\rho}\right)^m \frac{1}{2^m m!} \int e^{-i\omega t} \frac{K_m[(r/\rho)(1 - (i\omega/\gamma))]}{(1 - (i\omega/\gamma))^m} \frac{d\omega}{2\pi}, \tag{4.12}$$

where we have used eqn 2.12.4.28 of [13].

Since the value of $g$ can be varied continuously, corresponding powers of $g$ in equations (3.13) and (4.12) can be equated term by term. This yields

$$\int e^{-i\omega t} K_m\left[\frac{r}{\rho}\left(1 - \frac{i\omega}{\gamma}\right)\right] d\omega = \frac{\sqrt{\pi}}{2^m \Gamma(m + (1/2))} \gamma^{2m} e^{-\gamma t} \left(\frac{\rho}{r}\right)^m \left(t^2 - \frac{r^2}{v^2}\right)^{m-1/2}, \tag{4.13}$$

for $t > r/v$ and zero otherwise. The integral

$$\int e^{-i\omega t} K_m[\alpha - i\omega\beta]\, d\omega = \frac{\sqrt{\pi}}{2^m \Gamma(m + (1/2))} \frac{\alpha^m}{\beta^{2m}} e^{-\alpha t/\beta}(t^2 - \beta^2)^{m-1/2}, \tag{4.14}$$

is then obtained by writing $\alpha = r/\rho$ and $\beta = r/v$ so $r = \rho\alpha$, $v = \rho\alpha/\beta$ and $\gamma = \alpha/\beta$ on the right of equation (4.13). An extensive search has not found either of equations (4.13) or (4.14) in standard tables.

Taking the limit $\beta \to 0$ in equation (4.14) yields a delta function on the left. The result can then be rearranged to give

$$\delta(t) = \lim_{\beta \to 0} \frac{1}{2\pi K_m(\alpha)} \frac{\sqrt{\pi}}{2^m \Gamma(m + (1/2))} \frac{\alpha^m}{\beta^{2m}} e^{-\alpha t/\beta}(t^2 - \beta^2)^{m-1/2}. \tag{4.15}$$

Although this result is somewhat baroque, such re-expressions of the delta function as limiting forms can be useful in derivations.

# 5. Summary and conclusion

The propagator (or Green function or impulse response function) of cortical large-scale neural activity and small-scale haemodynamic responses has been investigated, leading to the exact analytic form in equation (3.18). This expression reproduces several prior results in limiting cases, improves on a recent approximation, and enables the evaluation of a number of related integrals and series that do not appear to be contained in standard tables. In applications, the results will be of use in analytic work on responses evoked by stimuli, neural avalanches and analysis of fMRI scans. Relationships to retarded propagators of relativistic scalar bosons and the telegrapher's equation also exist but the direct derivation here is both instructive and yields a variety of new results.

Data accessibility. This article has no additional data.

Competing interests. I declare I have no competing interests.

Funding. This work was supported by the Australian Research Council Center of Excellence grant no. CE140100007 and the Australian Research Council Laureate Fellowship grant no. FL1401000225.

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
