## [Peer Review File · Royal Society Open Science]

Review History

Decision letter (RSOS-211562.R0)

Dear Dr Robinson

On behalf of the Editors, we are pleased to inform you that your Manuscript RSOS-211562 "Integrals and Series Related to Propagators of Neural and Hemodynamic Waves" has been accepted for publication in Royal Society Open Science subject to minor revision in accordance with the referees' reports. Please find the referees' comments along with any feedback from the Editors below my signature.

Please submit your revised manuscript and required files (see below) no later than 7 days from today's (ie 28-Oct-2021) date. Note: the ScholarOne system will 'lock' if submission of the revision is attempted 7 or more days after the deadline. If you do not think you will be able to meet this deadline please contact the editorial office immediately.

on behalf of Dr Peter Stewart (Associate Editor) and Mark Chaplain (Subject Editor)
openscience@royalsociety.org

Associate Editor Comments to Author (Dr Peter Stewart):

Comments to the Author:

Dear Prof Robinson,

Having considered your revisions and responses to the original reviews for this manuscript, I am happy to recommend publication in Royal Society Open Science subject to a minor revision.

Section 2 (the model description) of the paper has a very high overlap with several previous papers (the iThenticate report shows a number of paragraphs have been lifted almost verbatim). While appreciating that there will always be some overlap, in such cases, this does seem rather extreme. Please revise this section to reduce the overlap - thank you.

Best wishes,
Peter

===PREPARING YOUR MANUSCRIPT===

You should provide two versions of this manuscript and both versions must be provided in an editable format
one version should clearly identify all the changes that have been made (for instance, in coloured highlight, in bold text, or tracked changes);
a 'clean' version of the new manuscript that incorporates the changes made, but does not highlight them. This version will be used for typesetting.

===PREPARING YOUR REVISION IN SCHOLARONE===

-- Ensure that your data access statement meets the requirements at <https://royalsociety.org/journals/authors/author-guidelines/#data>. You should ensure that you cite the dataset in your reference list. If you have deposited data etc in the Dryad repository, please only include the 'For publication' link at this stage. You should remove the 'For review' link.

-- If you are requesting an article processing charge waiver, you must select the relevant waiver option (if requesting a discretionary waiver, the form should have been uploaded, see 'File upload' above).

-- If you have uploaded any electronic supplementary (ESM) files, please ensure you follow the guidance at <https://royalsociety.org/journals/authors/author-guidelines/#supplementary-material> to include a suitable title and informative caption. An example of appropriate titling and captioning may be found at https://figshare.com/articles/Table_S2_from_Is_there_a_trade-off_between_peak_performance_and_performance_breadth_across_temperatures_for_aerobic_scope_in_teleost_fishes_/3843624.

Author's Response to Decision Letter for (RSOS-211562.R0)

See Appendix A.

Decision letter (RSOS-211562.R1)

Dear Dr Robinson,

I am pleased to inform you that your manuscript entitled "Integrals and Series Related to Propagators of Neural and Hemodynamic Waves" is now accepted for publication in Royal Society Open Science.

on behalf of Dr Peter Stewart (Associate Editor) and Mark Chaplain (Subject Editor)
openscience@royalsociety.org

Appendix A

Dear Dr Stewart,

Thanks for your decision letter on this MS.

Whatever your “iThenticate” software says, there is no “extreme” overlap in the one page of background material and neither referee raised this as an issue. This is the gross pitfall of relying on blind pattern matching software. Background material **should** overlap with descriptions of existing objects (e.g., the central integral here) being in the **same** notation and words as far as possible, to avoid ambiguity and confusion. Nothing in Sec. 2 was stated as being a new result; indeed, all material was clearly referenced as being background. Changing words for the sake of changing words may help sales of thesauruses but is not scientifically appropriate or required.

I have added an introductory paragraph to Sec. 2 to make it abundantly clear that the material is background. Despite the above, just to expedite matters, I have made some other cosmetic changes to Sec. 2 so iThenticate’s nonexistent mind is less disturbed – but I can’t change the content or its essential scientific features without rendering it incorrect or misleading.

I trust that this satisfies the remaining requirements.

Please copy any response emails to robinsonp5@bigpond.com as well as my official email as it seems that the University of Sydney blocks emails from your address.

Best regards,

Peter Robinson